# Structural insights into photoactivation of plant Cryptochrome-2

Malathy Palayam[1,2], Jagadeesan Ganapathy[1,2], Angelica M. Guercio[1,2], Lior Tal [1], Samuel L. Deck[1] & Nitzan Shabek [1✉]

Cryptochromes (CRYs) are evolutionarily conserved photoreceptors that mediate various light-induced responses in bacteria, plants, and animals. Plant cryptochromes govern a variety of critical growth and developmental processes including seed germination, flowering time and entrainment of the circadian clock. CRY's photocycle involves reduction of their flavin adenine dinucleotide (FAD)-bound chromophore, which is completely oxidized in the dark and semi to fully reduced in the light signaling-active state. Despite the progress in characterizing cryptochromes, important aspects of their photochemistry, regulation, and light-induced structural changes remain to be addressed. In this study, we determine the crystal structure of the photosensory domain of Arabidopsis CRY2 in a tetrameric active state. Systematic structure-based analyses of photo-activated and inactive plant CRYs elucidate distinct structural elements and critical residues that dynamically partake in photo-induced oligomerization. Our study offers an updated model of CRYs photoactivation mechanism as well as the mode of its regulation by interacting proteins.

[1] Department of Plant Biology, University of California – Davis, One shields Avenue, 1002 Life sciences, Davis, CA 95616, USA. [2]These authors contributed equally: Malathy Palayam, Jagadeesan Ganapathy, Angelica M. Guercio. ✉email: nshabek@ucdavis.edu

ryptochromes (CRYs) are evolutionarily conserved photoreceptors that regulate numerous developmental networks including de-etiolation, photoperiodic control of flowering, root growth, plant height, organ size, stomatal opening, and stress responses[1–5]. Plants encode three different CRYs (CRY1–3); CRY1 and CRY2 function in the nucleus and predominantly regulate seedling development and flowering time, while CRY3 mostly functions in subcellular organelles[5–7]. CRYs are comprised of two core domains: the highly conserved N-terminal photolyase homologous region (PHR) and the diversified carboxy-terminal extension (CCE). The PHR domain is further subdivided into N-terminal α/β and C-terminal α subdomains[8–10]. The chromophore, flavin adenine dinucleotide (FAD), is non-covalently bound to the C-terminal α subdomain of the PHR and plays a critical role in facilitating photoreduction[11–15]. Therefore, the PHR domain serves as the functional domain responsible for the light sensing mechanism of plant CRYs. Unlike the conserved PHR domain, the CCE domain is disordered, highly dynamic, and varies in length[16–18]. The CCE domain plays role in the regulation of CRYs by interacting with constitutive photomorphogenesis 1 (COP1) ubiquitin ligase as part of light signaling pathway[19–22].

FAD bound to plant CRYs–PHR can be found in distinct redox states: the fully oxidized FAD, the semireduced $FADH^\bullet$ or $FAD^{\bullet-}$, and the fully reduced $FADH^-$ or $FADH_2$[23]. Among the different redox forms, only the oxidized flavin and anion radical semiquinone flavin ($FAD^{\bullet-}$) can absorb blue light[15,23]. Oxidized FAD ($FAD_{ox}$) is thought to be the ground state chromophore of plant CRYs, since it absorbs blue light most efficiently and can be rapidly photoreduced[15,23,24]. Photoreduction of FAD is proposed to be carried out by electron transfer among three conserved tryptophan residues referred to as "Trp-triad" within the PHR domain[24–27]. Furthermore, in plant CRYs, the aspartic residue adjacent to the isoalloxazine ring of FAD coordinates the photocycle by acting as the FAD proton donor coupled to the Trp-based electron transfer chain[28,29]. Despite the biochemical data, the function of Trp residues in planta has been questioned after it was shown that photo-physiological activities of CRYs can be carried out even upon mutation of the Trp triad[30]. Hence, the photoactivation mechanism of CRYs and how the photocycle is able to trigger the conformational changes necessary for signal transduction remains elusive[11,12,31]. It has been largely accepted that blue light induces the homo-oligomerization of plant CRYs and the CRY oligomer represents the activated state, whereas the monomer exhibits the inactivate state[32–34]. Blue light-induced oligomerization of CRYs was also found to be regulated by protein interactions with various signaling partners such as bHLH inhibitor of cryptochrome (BIC)s and CIBs transcription factors[35,36]. Recently, the structure of the PHR domain of Arabidopsis CRY2 has been determined as a monomer as well as in a complex with BIC2[37]. These structures provide important insights into the mechanism by which the inhibitor BIC2 binds to and regulates cryptochromes. Additionally, the structure of bioactive Zea mays mutant CRY1 ($ZmCRY1C^{W374A}$) has been reported and proposes a mechanism for photoactivation by light-mediated dimerization or oligomerization in plant CRYs[19]. Despite the notable progress in characterizing and understanding the photo-oligomerization of CRYs at the structural level, a detailed crystal structure of native plant CRY2 in the oligomeric photoactivated state has yet to be determined.

Here, we report the crystal structure of the Arabidopsis CRY2 PHR domain in a tetrameric active state. Our systematic structure-based analyses of photoactivated and inactive BICs–CRYs has identified new structural elements and critical residues that dynamically partake in photoinduced oligomerization. Furthermore, comparative examination of CRYs conformation suggests

that the tetrameric CRY2 recapitulates an intermediate state of electron transport via the Trp-triad. Our study offers an important updated model of CRYs photoactivation mechanism as well as the mode of its regulation by interacting proteins.

## Results

### Crystal structure of tetrameric Arabidopsis CRY2–PHR.
To examine the oligomeric structure of photoactivated CRY2, we expressed, purified and crystalized the complete PHR domain of wild-type Arabidopsis CRY2. Following extensive crystallization trials under full spectrum light conditions, we successfully determined the crystal structure of Arabidopsis CRY2-PHR in a tetrameric state (denoted $AtCRY2\text{-}PHR_{tetamer}$). $AtCRY2\text{-}PHR_{tetramer}$ is comprised of four monomeric units (A, B, C, D) that are arranged as a ring-like structure with a central hollow cavity (Fig. 1a and Table 1). The tetrameric conformation of $AtCRY2\text{-}PHR_{tetramer}$ is related by two-fold symmetry with an overall size of $110.5^2$ Å (Fig. 1a). Each individual unit of the PHR domain is comprised of an N-terminal α/β domain (residues 5–132), C-terminal α domain (residues 214–487) and a connector loop (residues 133-213), which links the domains together (Fig. 1b and Supplementary Fig. 1). The co-factor, FAD, adopts a typical U-shaped conformation and is buried inside the central core of C-terminal α domain, similar to the conformation reported in other structures of cryptochromes and photolysases (Fig. 1c). Size exclusion and multi-laser light scattering (SEC-MALS) analyses along with steady state spectrum analysis suggest that $AtCRY2\text{-}PHR$ in solution is bound to oxidized FAD, displayed by a distinct yellow color, and mostly exists as monomers with relatively small fractions of oligomeric states (Supplementary Fig. 2a–c). While we were able to clearly identify the tetramer formation in solution under full spectrum light, we found that these oligomers tend to form large aggregates or photobodies, and therefore failed to separate at distinct UV-detectable levels (Supplementary Fig. 2a, b), as reported in early studies of cryptochromes[16,22,24,38,39]. The crystallization process and nucleation events under full spectrum light conditions appear to favor the tetrameric state. This is likely because the super-saturation events during vapor diffusion under full spectrum light conditions resulted in high concentration of photoinduced oligomerized CRY2. Further absorption spectroscopy analysis of the oligomerized CRY2 crystals suggests a reduced state of FAD bound to PHR, as exemplified by non-detectable absorption peaks compared to the resting state $CRY2\text{–}FAD_{ox}$ in solution (Supplementary Fig. 2c).

### Structure guided comparison analyses of AtCRY2-PHR indicates a photoactivated tetrameric state.
The architecture within tetrameric $AtCRY2\text{-}PHR$ is comprised of two different interfaces: a head to tail (H–T interface, monomers A–B) and head to head (H–H interface, monomers A–D) interaction between the monomers (Fig. 2a, b). Superposition of each of the individual monomers A through D reveals a similar overall structure with root mean square deviation (r.m.s.d) values ranging from 0.33 to 0.62 Å as measured for $C^\alpha$ atoms (Supplementary Fig. 3a). In the $AtCRY2\text{-}PHR_{tetramer}$, the interface (H–T) is assembled by the residues from α6, α12, α13, α18, α19, $3_{10}$, L-24, L-26 (Fig. 2a) and the second interface (H–H) is formed by the residues from α2, α10, α7, and L-11 (Fig. 2b). Analysis of the H–T interface reveals highly conserved residues that form salt-bridges and hydrogen bond interactions between the monomers (Fig. 2a and Supplementary Fig. 1). However, the H–H interface is stabilized by fewer ionic and hydrogen bonds interactions than the H–T (Fig. 2b). Structural superposition of the $AtCRY2\text{-}PHR_{tetramer}$ with the photo-active mutant $ZmCRY1C^{W368A}$ shows a similar structural

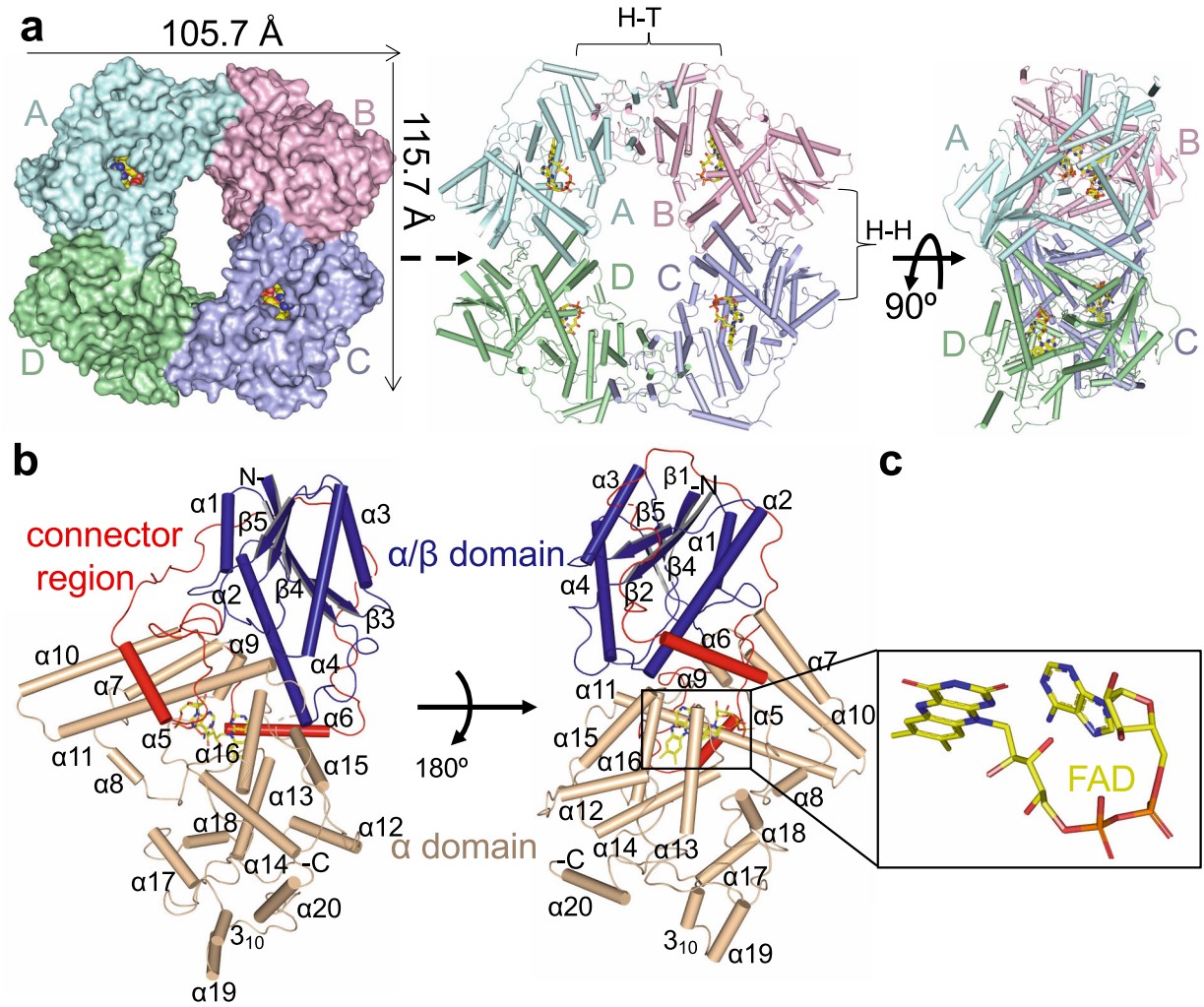

**Fig. 1 Molecular architecture of CRY2–PHR. a** Overall structure of CRY2–PHR bound to FAD shown in top view surface representation, and side view with the four monomers A–D assembled to a tetramer (A, B, C, D represented by cyan, pink, purple, and green respectively). FAD molecules are shown in yellow. Head to head and head to tail interfaces are denoted by brackets as H–H and H–T respectively. **b** Close-up view of monomer A of *At*CRY2-PHR$_{tetramer}$ represented in cartoon. Secondary structure of the indicated domains shown as cylinders for helices and beta strands and labeled in black. N-terminal α/β domain (blue), C-terminal α-domain (light brown), and flexible connector loop (red) are highlighted. **c** Close-up view of bound FAD represented in sticks and colored by elements: yellow (carbon), blue (nitrogen), red (oxygen), phosphate (orange).

arrangement yet a large r.m.s.d of 3.0 Å for 1368 residues (Supplementary Fig. 3b). The superposition of a single monomer from each of these oligomeric structures shows an almost identical fold of the Cα atoms (Supplementary Fig. 3c). The greater deviation when looking at the entirety of the tetramer is likely because of sequence variation between *Zea mays* CRY1 and *Arabidopsis thaliana* CRY2 and possibly species-specific variation of oligomeric states. Further structural analysis revealed the presence of a unique 3$_{10}$ helix in *At*CRY2-PHR$_{tetramer}$ involved in the active H–T interface. This unique 3$_{10}$ helix is present as a η-helix in active mutant of *Zm*CRY1C$^{W368A}$ and in the monomeric *At*CRY2-PHR (Supplementary Figs. 3b, 3d$_{(ii)}$). Next we compared *At*CRY2-PHR$_{tetramer}$ structure to the recently reported *At*CRY2-PHR monomeric structure (denoted *At*CRY2-PHR$_{monomer}$). Superposition analysis of *At*CRY2-PHR$_{tetramer}$ H–H (monomers A-D and B-C) with the two copies in asymmetrical unit of *At*CRY2-PHR$_{monomer}$ (represented as copies A′ and B′) reveals no major changes in the overall structure (Supplementary Fig. 3d$_{(i)}$). However, superposition of *At*CRY2-PHR$_{tetramer}$ H-T (monomers A-B) of *At*CRY2-PHR$_{tetramer}$ with the two copies of *At*CRY2-PHR$_{monomer}$ shows that only monomer A of the tetramer is aligned with the copy A′ of the *At*CRY2-PHR$_{monomer}$

(Supplementary Fig. 3d$_{(ii)}$). Similar examination of the crystallographic two-fold symmetry related copies of *At*CRY2-PHR$_{monomer}$ shows larger deviation in particularly within the helices that participate in the active dimerization interface (Supplementary Fig. 3d$_{(iii)}$). This analysis strongly suggests that the H–H interface region of *At*CRY2-PHR$_{tetramer}$ is found to be similar in *At*CRY2-PHR$_{monomer}$ and *At*CRY2-PHR$_{inactive}$ structures, however H–T interface in *At*CRY2-PHR$_{tetramer}$ is distinct during the active oligomeric state. Further comparison of the H–H interface shows fewer hydrogen bonds in *At*CRY2-PHR$_{tetramer}$ compared to *At*CRY2-PHR$_{monomer}$. This plasticity may result in larger conformational changes during the oligomerization process of CRY2, and can explain the movement of the subdomains that results in fewer interactions within the H–H interface. Also, *At*CRY2-PHR$_{monomer}$ and the photoactive mutant *Zm*CRY1C$^{W368A}$ have three η-helices located in the H–T interface that are completely absent in *At*CRY2-PHR$_{tetramer}$, suggesting more flexibility of the H–T interface and a possible role in fine-tuning the proper orientation of the dimeric interface within the active tetrameric structure (Supplementary Fig. 3d$_{(iii)}$). This analysis also further corroborates the recent findings that place the H–T interface as the initial interaction surface for inhibition

**Table 1 Data collection and refinement statistics.**

|  | AtCRY2$_t$ |
| --- | --- |
| *Data collection* |  |
| Space group | P 3$_1$ |
| *Cell dimensions* |  |
| a, b, c (Å) | 208.362, 208.362, 81.607 |
| α, β, γ (°) | 90, 90, 120 |
| Resolution (Å) | 50.05–3.25 (3.36–3.25) |
| $R_{sym}$ | 0.078 (0.940) |
| $I/\sigma I$ | 14.8 (2.2) |
| Completeness (%) | 99.34 (97.64) |
| Redundancy | 1.5 (1.4) |
| *Refinement* |  |
| Resolution (Å) | 3.25 |
| No. reflections | 61,839 |
| $R_{work}/R_{free}$ (%) | 27.1/32.9 |
| No. atoms | 15,611 |
| Protein | 15,279 |
| Ligand/ion | 302 |
| Water | 30 |
| *B*-factors | 53.80 |
| Protein | 54.04 |
| Ligand/ion | 44.87 |
| Water | 19.98 |
| *R.m.s. deviations* |  |
| Bond lengths (Å) | 0.005 |
| Bond angles (°) | 1.03 |
| PDB ID | 6X24 |

Values in parentheses are for highest-resolution shell.

of photo-activation by BIC2, and substantiates this interface as a unique feature of active tetrameric structures[19,35,36,40]. Moreover, the structural comparison of AtCRY2-PHR$_{tetramer}$ with AtCRY2-PHR$_{monomer}$ reveals approximately 54 structural alterations in amino acid side chain rotamers (Supplementary Fig. 3e). Interestingly, most of these changes occur in highly conserved polar residues and very few hydrophobic amino acids are altered. These differences in polar residue rotamer are likely to play a role in salt bridge formation, and stabilization of the active oligomeric state as was recently exemplified by mutational analysis[19,32]. One such change is the R439L and W349A mutation that leads to disruption of active dimer interface (H–T) formation both in vitro and in vivo, and reduced binding of CIB1 peptide to the CRY2 photoactive H–T interface[19]. Altogether, these analyses strongly suggest that AtCRY2-PHR$_{tetramer}$ represents a photoactivated tetramer and underlines the specific importance of the H–T interface in photo-oligomerization.

**The dynamic role of AtCRY2-PHR connecting loop in photo-oligomerization.** To further characterize CRY's structural conformation and photo-oligomerization states, we carried out structure-guided comparative analyses of active and inactive CRYs. We uncovered a dynamic, highly conserved interconnecting loop that wraps around the α/β domain (Supplementary Fig. 1). In the AtCRY2-PHR$_{tetramer}$ this connecting loop has two defined helices (α5 and α6) but it is largely disordered (Fig. 3a and Supplementary Fig. 1). As expected for active states, the conformation of the connecting loop within AtCRY2-PHR$_{tetramer}$ is found to be relatively similar to the

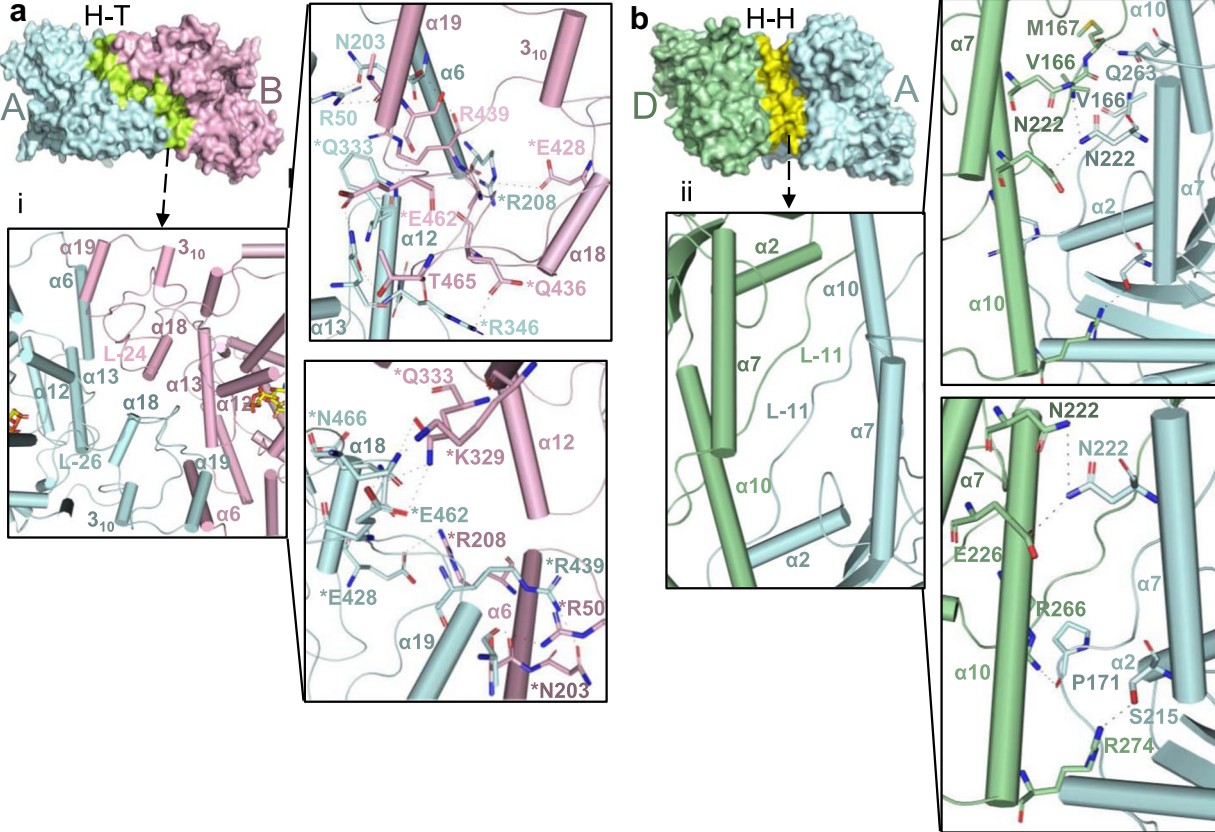

**Fig. 2 Oligomeric interfaces of AtCRY2-PHR. a** Side view surface representation of head to tail (H–T) interface (green) of A and B monomers (cyan and pink respectively); (i) Close-up views of H–T interface indicated by helices positions and all the amino acids that form direct interactions. Asterisks represent residues that form salt-bridges. **b** Side view surface representation of head-to-head (H–H) interface (yellow) of A and D monomers (cyan and green respectively); (ii) Close-up of H–H interface indicated by helices positions and all the amino acids that form direct interactions.

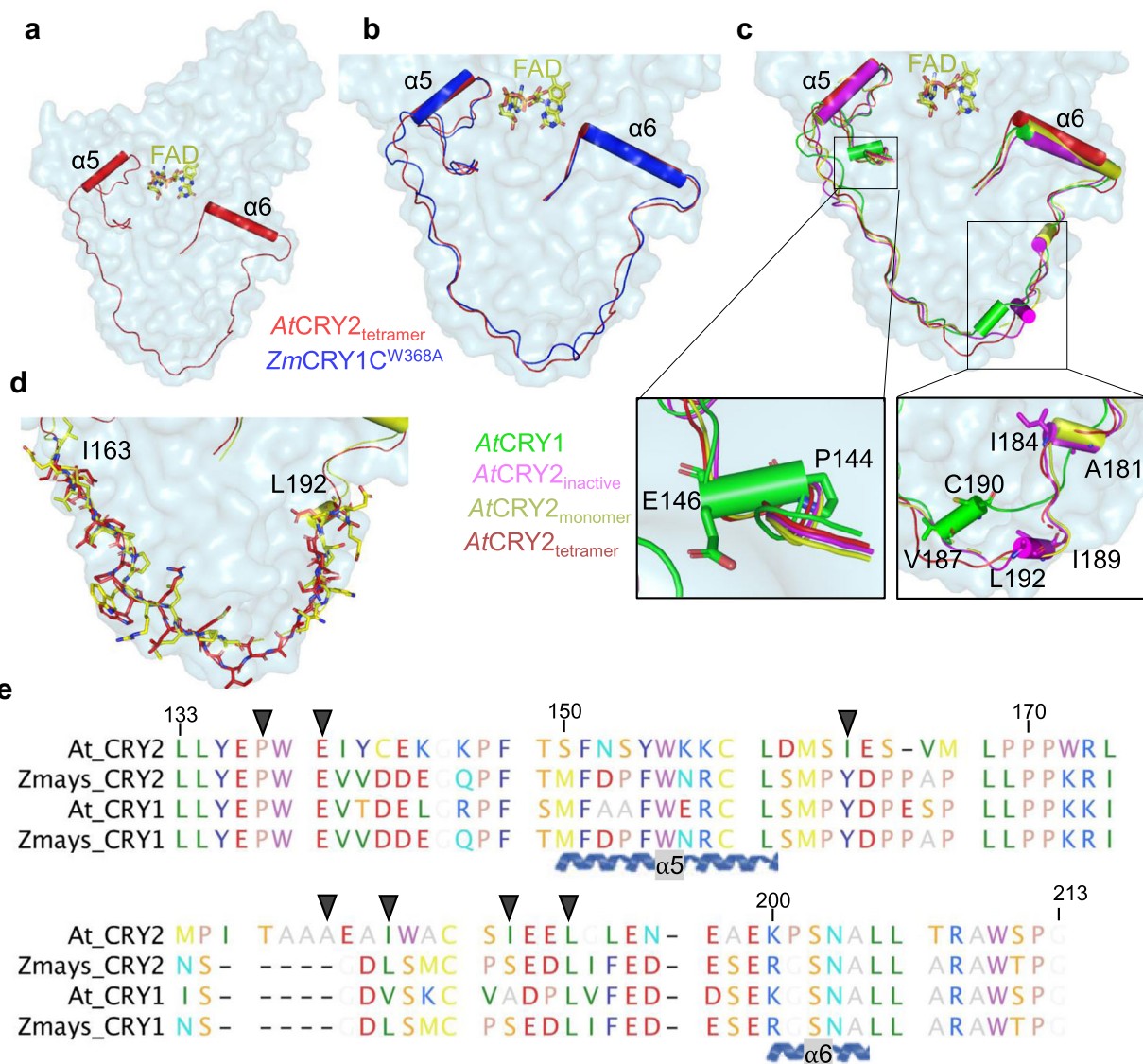

**Fig. 3 Structural and functional variation of the interconnecting loop. a** Surface representation of monomeric *At*CRY2-PHR$_{tetramer}$ (light blue) and connector region of PHR domain is highlighted in red. Helices are shown as cylinders. **b** Superposition of connector regions of *At*CRY2-PHR$_{tetramer}$ (red) and active mutant *Zm*CRY1C$^{W368A}$ (blue, PDB: 6LZ3); 0.8 Å r.m.s.d is calculated by PyMOL. **c** Comparison of connector region of *At*CRY2-PHR$_{tetramer}$ (red) with *At*CRY1 (green, PDB: 1U3C), *At*CRY2-PHR$_{monomer}$ (yellow, PDB: 6K8I), and *At*CRY2-PHR$_{inactive}$ (Magenta, PDB: 6K8K); r.m.s.d of 1.65 Å, 1.49 Å and 1.78 Å, respectively are calculated by PyMOL. **d** Sidechain view of connector loop region of *At*CRY2-PHR$_{tetramer}$ (red) and *At*CRY2-PHR$_{monomer}$ (yellow, PDB: 6K8I). **e** Sequence alignment and conservation of CRYs interconnecting loop. Residues are colored by traditional amino acid properties in RasMol colors. Arrows indicate residues highlighted in **a**–**d**.

*Zm*CRY1C$^{W368A}$ (Fig. 3b), suggesting a unique structural feature of photoactivated CRYs. Remarkably, comparative characterization of non-oligomeric CRYs reveals structural deviation between active and inactive states. We identified multiple secondary structure differences centered in 3$_{10}$ helix and short α-helices (residues 144–146, 181–184, 187–190, and 189–192, Fig. 3c). These structural variances strongly suggest a large movement of the N and C domains between the active and inactive states of CRYs (Fig. 3c and Supplementary Movie 1). Despite the sequence conservation in the connecting loop, superposition of the residues 163–190 in *At*CRY2-PHR$_{monomer}$ demonstrates major side chain alterations compared to *At*CRY2-PHR$_{tetramer}$ (Fig. 3d, e). These changes can be explained by the inherent flexibility of the unstructured coiled coil, and the overall structural alterations between active and inactive states of CRYs (Supplementary Movie 2). Notably, the disulfide bond C80–C190 in *At*CRY1-PHR

between the α/β domain and the connector region is not found in *At*CRY2-PHR, wherein the non-conserved residue C80 is replaced with D73 (Fig. 3e and Supplementary Fig. 1). Our data suggest that the absence of secondary structural elements in active CRYs considerably increases the plasticity, and the movement of α6 helix to actively participate in the interface formation. Furthermore, the connecting loop contains multiple residues that were already shown to be implicated in the photoactivated oligomeric interface such as R208, N203, and S202[19]. This further validates the flexibility of the connecting loop and places it as one of the central structural elements that regulates the photo-oligomeric state.

**Structural insights into inhibition of CRYs oligomerization by BICs.** To further delineate the mode of BIC2 inhibition of CRY2

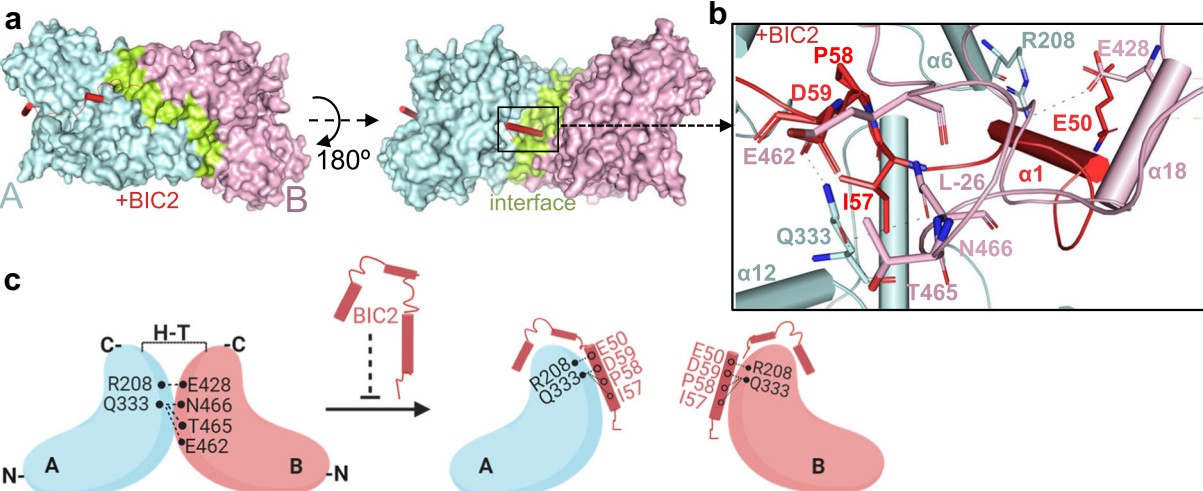

**Fig. 4 BIC2 disrupts oligomeric formation. a** Modeling of *At*CRY2-PHR$_{tetramer}$ complexed with *At*BIC2 (red) (superposition of *At*CRY2-PHR$_{tetramer}$ with *At*CRY2-PHR$_{inactive}$-BIC2 complex, PDB: 6K8K). Monomers A and B are colored in cyan and pink respectively. H–T interface is shown in green. **b** Close-up view of BIC2 (red) disrupting H–T interface. Amino acid residues are labeled and colored as indicated in **a**. **c** Proposed model of BIC2 inhibition (prepared by BioRender).

oligomerization, we examined the structural changes between active *At*CRY2-PHR$_{tetramer}$ and *At*CRY2-PHR$_{inactive}$ (inhibitory CRY–BIC2 complex)[37]. The H–T interface in *At*CRY2-PHR$_{tetramer}$ is formed by L-26 on one side of the monomer and α18 on the other side (Fig. 4a, b). In *At*CRY2-PHR$_{inactive}$ the α1 helix of BIC2 is inserted straight in the middle of the H-T interface and clearly clashes with the residues in α18, α19, and L-26, thus preventing or disrupting the H–T interface and oligomerization of *At*CRY2 (Fig. 4b). Specifically, within *At*CRY2-PHR$_{tetramer}$ H–T interface, the residue R208 (monomer A) forms ionic interactions with E428 (monomer B). Similarly, Q333 residue (monomer A) forms ionic interactions with E462 (monomer B), and bifurcated hydrogen bond interactions with T465 and N466 (monomer B). These interactions are completely disrupted upon BIC binding, where the residues E50 and D59 of BIC2 form ionic interactions with R208 and Q333 (Figs. 3 and 4b, c). Here, the *At*CRY2-PHR tetrameric structure provides a unique opportunity to inspect the specific mode of BIC2 oligomeric disruption. Our structural analysis corroborates certain residues involved in the CRY2–BIC2 interface as suggested previously[37], but also reveals new key conserved residues such as E462, T465, and N466 that are likely involved in oligomerization and the BIC2 mechanism of inhibition (Fig. 4b, c).

**Comparative analysis of FAD binding pocket of photo activated CRYs.** Given the critical role of FAD in the photoreduction process, we used the *At*CRY2-PHR tetrameric structure to provide a detailed analysis of FAD binding cavity in the photoactivated state. The FAD binding pocket within the *At*CRY2-PHR$_{tetramer}$ occupies a larger surface area of 849.6 Å$^2$ compared to *At*CRY2-PHR$_{monomer}$, which occupies a surface area of 821 Å$^2$. The FAD access cavity is comprised of positively charged residues that stabilize the negative-rich phosphate moiety of FAD (Fig. 5a). The adenine and isoalloxazine ring of FAD are stabilized by polar and negatively charged residues, that form the core region of the FAD pocket (Fig. 5b). The flexible phosphate moiety of FAD is held in place by forming hydrogen bonds with the backbone and side chains atoms of the residues T244, S245, L246, S248, W353, R359, D387, and D389. The isoalloxazine and adenine di-nucleotide ring position parallel to each other, and each ring is stabilized by interactions with nearby amino acid residues (such as N356, R359,

D387, D389, and D393). This extensive hydrogen bond network substantially supports the U-shaped conformation of FAD (Fig. 5b and Supplementary Fig. 4a–c). A striking feature of plant cryptochrome ancestors, photolyases, is the substitution of the amino acid N380 to D393[41–43], which is a highly conserved key position within the FAD binding pocket (Fig. 5c). Despite the notion that photolyases and cryptochromes share structural similarities, they are distinct in their physiological function as well as their bound FAD redox state[42,44]. The diverged D393 residue plays an important role in determining the redox property of FAD by acting as an acid or base, and carries a net negative charge that destabilizes FADH$^-$ [41]. It has been shown that D393A mutants exhibit loss in the photoreduction activity[37]. Similarly, a mutant in *At*CRY1 of D396C leads to a complete loss or block of proton transfer to the N5 atom of Flavin[45]. Thus, N380 (photolyases) to D393 (CRYs) substitution may explain the difference between the redox states of bound FADH and FAD$_{ox}$ respectively. Interestingly, a comparison between plant CRY1 and CRY2 identified a key conserved change of D359 (CRY1s) to N356 in CRY2s; this conserved change is similarly found in other photolyases (Fig. 5c and Supplementary Figs. 1, 4d). In CRY1, D359 forms a strong hydrogen bond interaction with the oxygen atom of FAD, however in CRY2, N356 only forms electrostatic interactions. D359 may play an additional role in donating a proton to the flavin moiety as was reported previously[46], and both D359 (CRY1s) and N356 (CRY2s) are positioned within the ATP bindings site of CRYs[37,47,48]. The importance of this substitution and its potential involvement in the distinct photosensitivity that was reported between CRY1 and CRY2[11,49], await further research. Comparative structural analysis of the FAD cavity between *At*CRY2-PHR$_{tetramer}$ and *At*CRY2-PHR$_{monomer}$ shows almost no variations in the side chains (Supplementary Fig. 4c). Altogether, the absence of major deviations within the FAD binding pocket suggests that FAD cavity does not play a direct role in triggering conformational changes during oligomerization. Nonetheless, the distance between D393 and FAD in inactive CRYs (e.g., *At*CRY2-PHR$_{inactive}$) shows a larger difference compared to the active oligomeric structures (e.g., *At*CRY2-PHR$_{tetramer}$, Fig. 5d). This suggests that electron transfer related residues may undergo conformational changes during oligomerization and indirectly affect the FAD redox state.

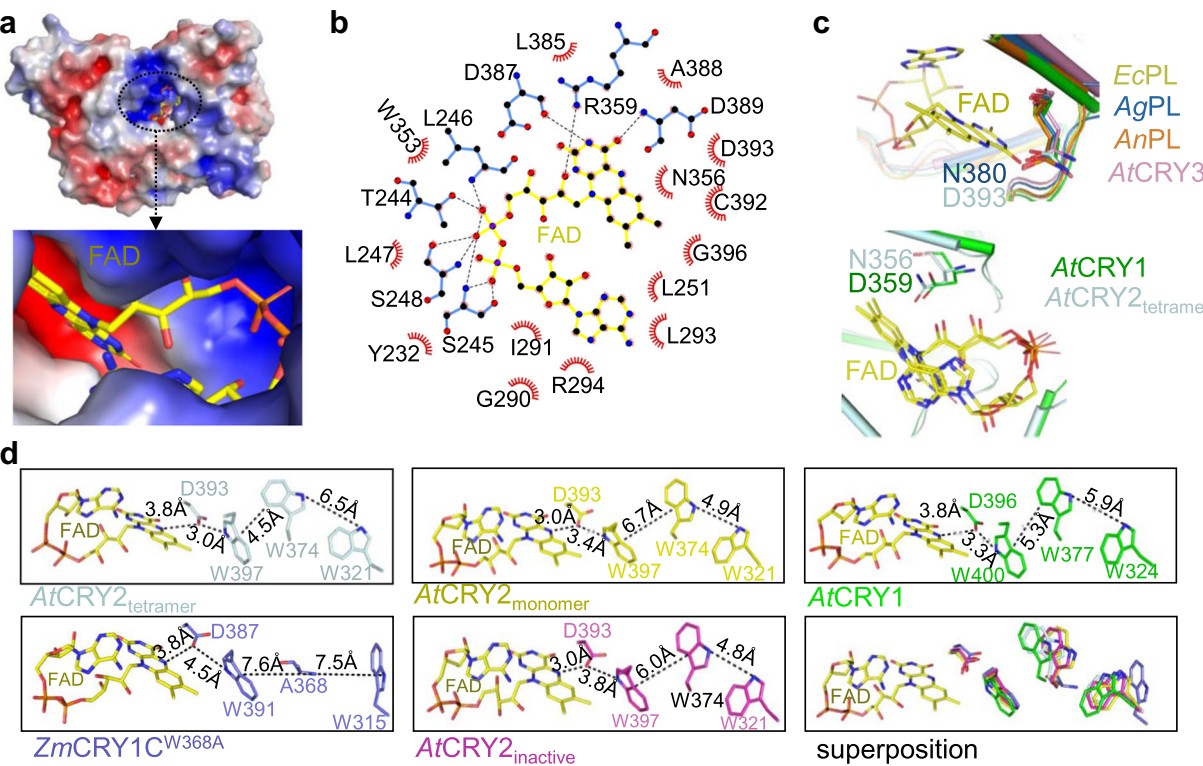

**Fig. 5 Structural analyses of FAD binding pocket and tryptophan (Trp) triad. a** Electrostatic surface representation of *At*CRY2-PHR_tetramer and close up of FAD (yellow) binding cavity. Electrostatic potential is color coded on the surface with red and blue representing areas of negative and positive charges, respectively. **b** Interactions of *At*CRY2-PHR_tetramer with FAD. FAD in yellow is colored with carbon (black), nitrogen (blue) and oxygen (red). Dotted lines in black represent hydrogen bonds between *At*CRY2-PHR_tetramer and FAD. Hydrophobic interactions are shown as spoked arcs (red) (prepared by LigPlot). **c** Close up view of FAD (yellow) and diverged residues within the binding pocket of *At*CRY2-PHR_tetramer (D393, light blue) superposition with *At*CRY1 (D396, green, PDB: 1U3C), *Arabidopsis*, *At*CRY3 (N428, Magenta, PDB:2VTB), *Agrobacterium tumefaciens* Photolyase, *Ag*PL (N380, blue, PDB:4U63)[41], *Escherichia coli* Photolyase, *Ec*PL (N378, Yellow, PDB: 1DNP)[64], and *Anacystis nidulans* Photolyase *An*PL (N386, orange, PDB:1TEZ)[65]. **d** Comparative analysis of plant cryptochromes tryptophan triad positions in relation to FAD. Residues are colored as in **c**. Distances were measured with PyMOL and indicated in dotted lines.

*At*CRY2-PHR tetrameric crystal structure captures an intermediate electron transfer via Trp-triad and implies conformational changes between active and inactive states. To better understand the role of the Trp-triad in electron transfer for the FAD and in photo-oligomeric states, we next carried out a systematic structural inspection of all reported CRYs structures including *At*CRY2-PHR_tetramer reported in this study. It has been previously suggested that the distances between tryptophan residues play an imperative role in the photocycle of cryptochromes, and determine their active state in vitro[20,50]. In the crystal structure of *At*CRY2-PHR_tetramer, the three key tryptophan residues: distal W321, central W374, and FAD-proximal W397 are aligned with the overall arrangement of the canonical Trp-triad (Supplementary Fig. 4e). Our comparative analysis found only subtle structural variation in the FAD-proximal tryptophan W397 of *At*CRY2-PHR_tetramer, *At*CRY2-PHR_monomer and *At*CRY2-PHR_inactive. However, remarkable distance deviations were measured between central W374 and distal W321 of *At*CRY2-PHR_tetramer and found to be greater (6.5 Å) compared to the distance measured for *At*CRY2-PHR_monomer (4.9 Å), AtCRY2-PHR_inactive (4.8 Å), and *At*CRY1-PHR (5.9 Å) (Fig. 5d). Interestingly, the distances between W374–W397 and W397–D393 are found to be shorter in *At*CRY2-PHR_tetramer (4.5 and 3 Å) and greater in *At*CRY2-PHR_monomer (6.7 and 3.4 Å), *At*CRY2-PHR_inactive (6 and 3.8 Å), *At*CRY1-PHR (5.3 and 3.3 Å), and *Zm*CRY1C (7.6 and 4.5 Å) (Fig. 5d and Supplementary Fig. 4e). The greater distances observed in *At*CRY2-PHR_monomer, *At*CRY2-PHR_inactive, and *Zm*CRY1C$^{W368A}$ from the FAD and within the Trp triad, indicate that the conformation adopted by tryptophan may not be suitable for efficient electron transfer in the inactive state. Altogether these results suggest that the Trp triad is rearranged upon exposure to blue-light to carry out active photoreduction process. Further analyses show that in *At*CRY2-PHR_monomer and *At*CRY2-PHR_inactive the distance between the D393 and N5 of isoalloxazine ring of FAD (O—H$^{D393}$...N5$^{ISO}$) complex is close (3.0 Å); unlike the active oligomeric structures *At*CRY2-PHR_tetramer and *Zm*CRY1C$^{W368A}$, where the distance between the D393 and N5 of isoalloxazine is measured to be farther (3.8 Å) (Fig. 5d). This is likely because in the active oligomeric state, D393 appears to be oriented towards W397 via hydrogen bonds. In comparison to monomeric CRYs, *At*CRY2-PHR_tetramer exhibits the largest distance between W321–W374, and the least distance between W374–W397 and W397–D393. This finding strongly suggests that *At*CRY2-PHR_tetramer is in a suitable active conformation for efficient electron transfer during photoreduction process. Altogether, our structure-based distance examination of the Trp triad and D393–W397 suggests that the crystal structure of *At*CRY2-PHR_tetramer has been trapped in a light-activated intermediate state, where the D393 accepts an electron from N1 of W397 and FAD is likely held between semi to fully reduced state (Fig. 5d and Supplementary Figs. 2c, 4e).

## Discussion

Since the discovery of plant CRYs, an increasing number of studies have provided an important insights that link light-

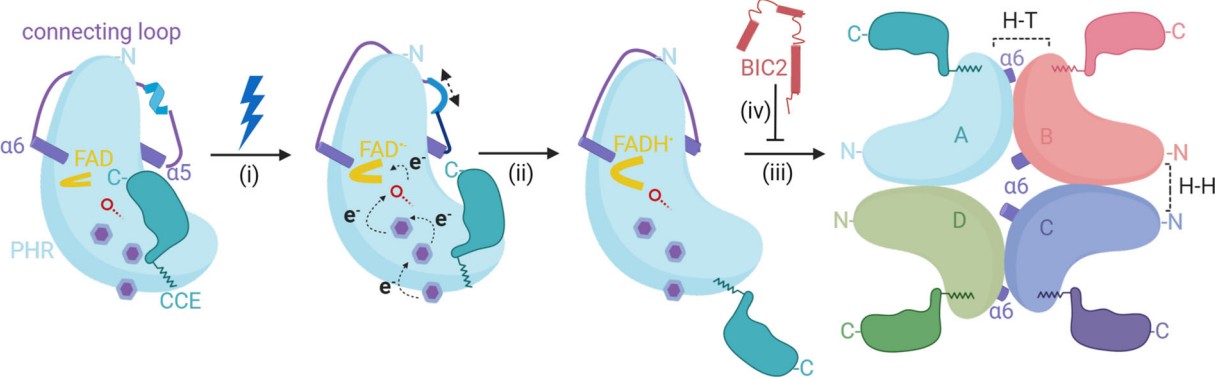

**Fig. 6 Proposed model of blue light-induced photoactivation and inactivation of plant CRYs.** CRYs exist as a monomer non-covalently linked to the oxidized FAD with the CCE tail bound to the PHR domain. The dynamic interconnecting loop in the inactive monomeric state is highlighted in purple with a short α-helix in light blue. (i) Blue light illumination causes the FAD to absorb energy and accept electrons via the tryptophan triad. The purple benzene rings represent the tryptophan residues and the red line with an open circle represents amino acid D393. After accepting the electrons, the FAD becomes FAD•− and adopts a U-shaped conformation followed by an overall change in the conformation of CRY including alterations in the interconnecting loop. (ii) The FAD•− becomes neutralized by accepting one proton from D393 and becomes FADH•. The overall change in the conformation of the CRY structure induced by photoactivation leads to the loss of secondary structural elements in the interconnecting loop. (iii) The photoactivated CRYs adopts an open conformation with CCE tail projecting outward leading to formation of homo-tetramer. The formation of the oligomeric state is guided by the dynamic interconnecting loop that allows the movement of residues in α-6 to participate in forming active interface in the oligomeric state. (iv) The photoactivated CRYs are readily inhibited by the presence of BICs which disrupts or prevents the H–T interface in the homo-tetramer.

induced photoreduction and activation of CRYs by homo-oligomerization[32,33,51]. The reported structure of mutant maize CRY1 ($Zm$CRY1C[W368A]) validates these functional studies by providing the photoactivated oligomeric states of CRYs[19]. Additionally, the crystal structure of the Arabidopsis inhibitory BIC2–CRY2 complex illuminated the mechanism of inactivation of CRYs to their monomeric form[37]. These studies advance our knowledge of photoresponsive events between activated CRYs and their desensitization upon binding with the dynamic regulatory inhibitor BICs. Nonetheless, our understanding of CRYs light signaling mechanism as well as the structural aspects required for their activation and oligomerization remained to be fully addressed. The tetrameric structure of native CRY2 had not been resolved in high resolution and a detailed structural analysis, that links photo-oligomerization and inhibition by BICs had yet to be performed. While there are many open questions regarding the photobiochemistry of these receptors in vivo, the finding that blue-light activates CRYs oligomerization has been established biochemically and *in planta*[19,32,37,40,52]. In this work, we report the active tetrameric crystal structure of the $At$CRY2-PHR. Our structural analysis dissects the active interface that is disrupted when BIC2 is bound. BIC2 represents a family of key transcriptional regulators including BIC1 and CIB1, therefore our proposed model of oligomeric inhibition as well as the residues involved are likely to be similar for other CRY inhibitory complexes. We defined two distinct interfaces in our structure, H–H and H–T, that are suggested to play roles in light-induced conformational changes required for photo-oligomerization. This analysis further corroborates the recent physiological and biochemical data of dimerization-deficient CRY2 mutants that had reduced binding affinity for $At$CIB1, and were unable to be degraded in blue-light-dependent manner[19]. Detailed structure-based analyses using the structure of $At$CRY2-PHR_tetramer allowed us to corroborate the recently reported CRY2 and CRY1 structures[19,37], and to further reveal residues that are involved in CRYs oligomerization. The newly identified residues E462 and N466 are highly conserved among all CRYs, and T465 in CRY2_tetramer is substituted to serine residue in all CRY1s (Supplementary Fig. 1).

Our findings reveal remarkable changes in approximately 54 amino acid orientations between the putative photoactive and inactive states of CRY2, suggesting that CRY2 undergoes conformational changes upon photoactivation (Supplementary Table 1). This study further illuminates a critical large evolutionarily conserved interconnecting loop (residues 133–213), that exhibits dynamic conformational changes between the active and inactive CRYs states. Several residues within the connecting loop were found to be directly implicated in photo-oligomerization (such as R208, S202, N203), and certain regions (such as α6) were found to be more amenable to the tetrameric state. Interestingly, the changes in the secondary structures between the active and inactive CRYs that are centered in the connecting loop suggest further molecular regulations. In fact, recent in vivo studies of photoexcited Arabidopsis CRY2 identified four key residues (T149, S162, T178, and S188) that undergo phosphorylation upon photoexcitation[53]. Notably, these residues are positioned within the dynamic short helices of CRYs (T178 and S188) or in close proximity to them, although defining the role of CRYs connecting loop in light-induced regulations awaits further studies.

The importance of the Trp-triad in CRYs and their role in photo-signaling has been addressed in many studies and has been a subject of debate in the field. We address the role of the Trp-triad by analyzing the structures of oligomeric active and monomeric inactive CRYs. Interestingly, within the Trp-triad we found distinct intraresidue distances between inactive and photoactive CRYs. Our detailed structure-based analysis suggests that the tryptophan residues likely adopt various structural rearrangements, that dictate the efficiency of electron transfer based on CRYs inactive or photoactive state. The calculated distances suggest that the oligomeric active CRY2 structure is likely recapitulating an activated intermediate electron transfer state between the FAD and D393–W397. This finding substantiates a number of mutational analysis studies that highlight the importance of these residues in preserving the redox potential and stabilization of the FAD radical in plant CRYs[29,30]. Although both photoactive CRY structures share a similar N5(FAD)-D393 distance and relatively shorter D393–W397 distance, the precise photoreduction mechanism of the W368A mutant that was used

in our comparative analysis, remains to be fully elucidated. Previous studies in vitro have shown that W374A mutation in CRY2 results in only partial photoreduction activity which may explain the larger Trp-triad distances found in our analysis[27,54]. Yet, *in planta* W374A CRY2 mutant results in a constitutively photoactive phenotype, thus it is also possible that alternative photoreduction pathway(s) can contribute to the photoactivated states as suggested previously[27,54].

Based on our current understanding including the crystal structure reported here, we propose the following model of CRY-mediated blue-light-induced photoreduction in plants (Fig. 6). Under dark conditions, CRYs are likely to exist as monomers, wherein the PHR surface is presumably bound with the CCE domain. Upon exposure to blue light, the FAD molecule absorbs energy and accepts electrons through the Trp triad photoreduction pathway (i). This would subsequently cause a structural rearrangement including the disassociation of the CCE domain (ii) from the PHR[9,16–18,55]. Because of the intrinsically disordered nature of the CCE domain, full length CRY1/2 structure has yet to be determined. The suggested photoinduced disassociation of CCE is based on several observations. For example, an in vitro proteolysis trypsin assay showed that the CCE domain of CRY1 was more susceptible to proteolysis in response to light[56]. Also, studies *in planta* have shown that blue-light-induced CCE phosphorylation of CRY2 resulted in electrostatic repulsion of the CCE domain from the PHR surface[16–18]. The resulting light-induced CCE rearrangement is likely to play a role in protein–protein interactions with downstream signaling partners[9,55,57], yet the exact function of CRYs' light-induced structural plasticity remained to be fully elucidated. In our proposed model, the photoexcited PHR domain undergoes homo-oligomerization, (iii) that involves multiple structural rearrangements including the conformational changes we observed in the interconnecting loop. These active oligomers are subsequently regulated by numerous downstream biological processes, (iv) that are responsible for the plant physiological response. By leveraging the reported CRY structures with our active tetrameric CRY2, we were also able to provide a comprehensive molecular view into BICs direct mode of inhibition. Whether BICs compete off the H–T interface or completely prevent oligomerization remains to be elucidated in vivo.

In summary, our study further extends the understanding of plant photosensing molecular mechanisms by revealing and analyzing the distinct structural changes between photoactive tetrameric and inactive monomeric states of cryptochromes.

## Methods

**Protein preparation and purification**. The CRY2–PHR (*Arabidopsis thaliana*, 1–498) was cloned into pFastbac-GST vector and expressed as a GST fusion protein in Hi5 suspension insect cells. CRY2–PHR was isolated from the soluble cell lysate by glutathione sepharose (GE Healthcare) using a buffer containing 50 mM Tris-HCl, pH 7.5, 200 mM NaCl, 4% Glycerol, 5 mM DTT. Proteins were further purified via on-column cleavage by TEV protease, followed by cation exchange and size exclusion chromatography. Protein was concentrated by ultra-filtration to 3–10 mg/mL$^{-1}$. Purified CRY2-PHR was eluted off a Superdex-200 gel filtration column (GE healthcare) in 20 mM Tris, pH 8.0, 200 mM NaCl, 2 mM DTT, 4% Glycerol. *At*CRY2-PHR protein existed mainly as monomer with an estimated molecular weight of 55 kDa, dimers and tetramers only accounted for a small portion of gel filtration elutions.

**Crystallization, data collection, and structure determination**. The crystals of *At*CRY2-PHR were grown at 25 °C by the hanging-drop vapor diffusion method with 1.0 μL protein complex sample mixed with an equal volume of reservoir solution containing 1.5 M sodium nitrate; 0.1 M BIS–TRIS propane pH 7.0. Hanging-drops crystallization trials were set up under continuous full-spectrum white light at 40 μmol m$^2$ s$^{-1}$ illumination. Crystals of maximum size were obtained and harvested from the reservoir solution with additional, 20% glycerol, serving as cryoprotectant. The crystals diffracted to ~3.2 Å resolution and the collected data were integrated, scaled with HKL2000 package[58]. The structure

solution of *At*CRY2-PHR$_{tetramer}$ was determined by molecular replacement using the *At*CRY1 structure (PDB: 1U3C)[59] as the search model. The structure model was manually built, refined, and rebuilt with PHENIX[60] and COOT[61].

**Size exclusion chromatography and multiangle-laser light scattering (MALS) analysis**. Purified proteins (20–50 μM) were injected onto Superdex-200 Increase 10/300 column (GE Healthcare) for analysis at a flow rate of 0.5 mL min$^{-1}$. The elution fractions (0.5 mL/fraction) were resolved by SDS-PAGE and analyzed by Coomassie Brilliant Blue G-250 stain. Molecular weights were estimated by comparison to known standards (Bio-Rad). For SEC-MALS analysis, Superdex-200 Increase 10/300 column was equilibrated with 50 mM HEPES pH 7.0, 120 mM KCl buffer (filtered through 0.22 μm filter and sonicated), recirculated through the system overnight at 0.5 mL min$^{-1}$. Hundred microliter of protein solution (2–3 mg mL$^{-1}$) was injected, and the data from detectors was exported at room temperature. All experiments were repeated thrice. Data was analyzed with ASTRA software package version 5.3.2.10 (Wyatt Technologies).

**UV–Vis protein absorption spectra**. Solution of purified *At*CRY2-PHR (2–3 mg mL$^{-1}$) absorption spectrum was measured on a NanoDrop One microvolume UV–Vis Spectrophotometer from 300–700 nm every 10 nm in triplicate under room light conditions and averaged. For absorption spectroscopy of *At*CRY2-PHR crystals, ~30 crystals were harvested into 1 μL of the crystal reservoir solution and measured similarly to the solution of purified *At*CRY2-PHR. Arbitrary absorption units (a.u.) were normalized against respective blank buffer conditions without protein or crystals. All experiments were repeated thrice.

**In silico sequence and structural analyses**. Representative cryptochrome sequences were gathered from NCBI protein database. Amino acid alignments were performed with CLC Genomics Workbench v12 using a slow progressive alignment algorithm. Photolyase sequence is included as an outgroup to plant cryptochromes.

The surface area of FAD binding pocket is analyzed using PDBePISA online tool[31]. For electrostatic charge distribution, the Adaptive Poisson–Boltzmann Solver (APBS) program[62] implemented in PyMOL was employed to calculate the electrostatic charge distribution of *At*CRY2-PHR$_{tetramer}$. All the figures in this manuscript was made using PyMOL. The LigPlot program[63] was used for 2-D representation of protein-ligand interactions in standard PDB data format.

**Statistics and reproducibility**. Data collection and refinement statistics of the crystal structure is provided in Table 1. All experiments described here were performed thrice.

**Reporting summary**. Further information on research design is available in the Nature Research Reporting Summary linked to this article.

## Data availability

The atomic coordinates of *At*CRY2-PHR structure has been deposited in the Protein Data Bank with accession codes 6X24. All relevant data are available from corresponding author upon request.

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

## Acknowledgements

This work is supported by UC Davis new faculty start-up funds. This research used resources of the Advanced Light Source, a U.S. DOE Office of Science User Facility under contract no. DE-AC02-05CH11231 that is supported in part by the ALS-ENABLE

program funded by the National Institutes of Health, National Institute of General Medical Sciences, grant P30 GM124169-01. We thank Ning Zheng and Jawdat Al-Bassam for discussion and help.

## Author contributions

N.S., M.P., J.G., and A.G. conceived and designed the experiments. N.S. and S.D. conducted the protein purification and crystallization experiments. N.S., M.P., and J.G. determined and analyzed the structures. N.S. and S.D. conceived and conducted the SEC-MALS, and spectroscopy experiments. M.P., J.G., and A.G. conducted in silico studies and analyses. N.S., M.P., J.G., A.G., and L.T., wrote the manuscript with the help from all other co-authors.

## Competing interests

The authors declare no competing interests.
