## [Peer Review File · Communications Biology]

This manuscript has been previously reviewed at another Nature Research journal. This document only contains reviewer comments and rebuttal letters for versions considered at Communications Biology.

Reviewers' comments:

Reviewer #2 (Remarks to the Author):

The authors have addressed all my comments for the previous submission, and it is ready to be published without further delay.

The present ms presents a much better structural analysis in comparison to the two ms that were published a few months ago in NSB (also reviewed by this referee). For example, one of the two NSB paper analyzed W374A mutant that we reported many years ago, which represented a significant progress but the wild-type CRY2-PHR successfully analyzed by this allow us to see the "real" structure without potential artifacts derived from the mutation. The second NSB paper analyzed the wild-type CRY2-PHR but without study of the tetramer structure. Moreover, the present study reveals more photoresponsive structural changes not previously reported, which is arguably the most important structural knowledge for a photoreceptor protein.

Reviewer #3 (Remarks to the Author):

The authors adequately addressed some of the concerns that I raised based on the manuscript originally submitted to Nature Communications. However, I still feel that the conclusions about the CRY photoactivation mechanism, that are drawn based on the presented structure of the light-state *Arabidopsis thaliana* (At) CRY2 tetramer structure in absence of any functional validation, are rather limited.

This reviewer thinks, that the main value/novelty of the At-CRY2-PHR structure compared to the already published tetrameric bioactive *Zea mays* CRY1 mutant structure (Shao et al, NSMB 2020, Ref.19) is that it allows a direct comparison of active At-CRY2-PHR with the published (inactive) monomeric At-CRY2-PHR and BIC-inhibited At-CRY2-PHR (Ma et al, NSMB 2020, Ref 37). I.e. potential differences between the activation mechanism of tetrameric *Zea mays* (Zm) CRY1 and At-CRY2 can be worked out and are now no-longer affecting the comparison between active and inactive CRY2. This is in my opinion not sufficiently clear in the current manuscript. The authors cite the paper reporting the tetrameric Zm-CRY1(W374A) structure for the "proposed photoactivation mechanism in CRYs (intro line 87 to 89)", but they do not say, what mechanism was proposed for CRY1 and how this differs from the mechanism proposed in this manuscript for At-CRY2. What specifically is new in this manuscript regarding the CRY2 photoactivation mechanism compared to what we already know from the tetrameric Zm-CRY1 structure? The authors partly address this issue structurally in the text and by adding a new table S1, that lists structural differences to the inactive At-CRY2 structures as well as to the active Zm-CRY1 structure, but the functional/mechanistic comparison between At-CRY2 and Zm-CRY1 is still largely missing.

Here are some examples, where the structural comparison needs more work:

Tetramer Architecture. Lane 140 – 149, 163:

The CRY2-PHR tetramer has the same architecture as the recently published bioactive tetrameric *Zea mays* CRY1. The 3 Å rmsd deviation that the authors report, appears to be due to the rotation of the C/D molecules compared to A/B (which the authors don't mention, but fig. S3B suggests). Minor differences are observed in the H-T tetramer interface, i.e. a unique 310 helix is not present in Zm-CRY1 and inactive CRY2, instead these have three η -helices. Unfortunately no functional implications of these changes apart from higher flexibility in the CRY2 tetramer are discussed.

Interfaces, Lane 149 to 163

The authors compare the CRY2 H-H and CRY2 H-T interfaces of the tetramer with the contacts of monomeric AtCRY2 to its crystallographic symmetry mates. The argumentation and Fig. S3d/e are a bit confusing, but it seems they want to say that the H-T interface in the At-CRY2 tetramer differs more from symmetry mate contacts in the At-CRY2 monomer crystals than the H-H interface. This implies that a similar tetramer was observed in the At-CRY2 monomer crystals by symmetry relationships. The authors should give more background information on the At-CRY2 monomer crystals (space group, symmetry relations etc.), if they want to do this comparison. Additionally, I find a comparison with crystal dimer interfaces (which are likely influenced by crystal contacts) critical for drawing functional conclusions.

Rotamers in Interfaces, Lane 169:

The authors describe 54 rotamer changes between the CRY2-tetramer and monomer (Fig. S3f). Can all of these rotamer changes be seen based on an electron density obtained at 3.25 Å resolution? For some of them (e.g. R439, R208) I doubt it. Please provide electron density figures to support these rotamer changes.

Connector Loop, Lane 180 onwards:

The authors describe conformational changes in the connector loop between the active tetrameric AtCRY2 and ZmCRY1 structures compared to the inactive monomeric CRY2 and BIC inhibited CRY2 structures and report enhanced flexibility in the active structures.

They suggest movement of the N- and C-terminal CRY domain (lane 191), which is not obvious based on the presented structural comparisons. Please clarify.

Moreover, conformational heterogeneity in the connector loop is frequently observed between different CRYs, as it is a flexible and dynamic region, and the significance of these changes for photoactivation is not clear based on the current manuscript. Please clarify.

Inhibition of CRYs by BIC (from lane 207)

The authors compare the H-T dimerization interface in the At-CRY2 tetramer with competing CRY2-BIC interactions. Was the competition of BIC with the H-T tetramer interface not evident based on the Zm-CRY1 tetramer structure? What is new based on the At-CRY2 tetramer structure?

Lane 222: The At-CRY2 structure "reveals key residues E462, T465, N466...." The authors should say if these residues conserved in Zm-CRY1 tetramer or different?

FAD binding pocket:

Lane 252: "D359 (ZmCRY1) vs N345 (At-CRY2) explain reported differences in light sensing mechanism between CRY1 and CRY2." The authors should explain which "differences in light sensing mechanism between CRY1 and CRY2" exactly they are addressing and provide a mechanistic rationale why they think that D359 vs. N345 affects exactly these light sensing differences.

Lane 256/257: 180° rotation in FAD O3 and O4. I doubt that this can be derived from a 3.25 Å resolution map. The authors should show this region with electron density, to show how the electron density defines this 180° rotation.

Trp triad:

Lane 275: "atomic resolution" is a bit bold for only 3.25 Å resolution.

The authors find that the distance between the middle W374 and the distal W321 is larger (6.5 Å) in the active At-CRY2 than in inactive AtCRY2 monomer (4.8/4.9 Å), and also somewhat larger than in the also tetrameric active Zm-CRY1 (5.9 Å).

The distances W374-W397 and W397-D393 are shorter in the active At-CRY2 tetramer compared to inactive At-CRY2 and also compared to active Zm-CRY1. This is confusing, as here the active Zm-CRY1

resembles inactive At-CRY2 and is different from active At-CRY2. The authors should explain this. D393 – N5 FAD: here active Zm-CRY1 and At-CRY2 both have shorter distances than inactive At-CRY2. Here tetrameric active Zm-CRY1 seems more similar to tetrameric active At-CRY2.

Is the Trp triad and D393 arrangement of tetrameric Zm-CRY1 more similar to tetrameric active At-CRY2 or to inactive At-CRY2 ? Please clarify this and potentially discuss mechanistic differences for the active CRY1 and active CRY2 state.

Overall the presented structural differences between active/inactive CRY2 and active CRY1 do not convincingly show, which features are characteristic of photoactivated At-CRY2 and why. The discussion unfortunately only recapitulates the observed structural changes, but also does not discuss clear mechanistic implications for light-activation derived from these observations.

I therefore think that this manuscript is more suited for a journal that is specialized on structural biology.

Point-to-Point Responses to Reviewers' Comments

Reviewer #2 (Remarks to the Author):

The authors have addressed all my comments for the previous submission, and it is ready to be published without further delay.

The present ms presents a much better structural analysis in comparison to the two ms that were published a few months ago in NSB (also reviewed by this referee). For example, one of the two NSB paper analyzed W374A mutant that we reported many years ago, which represented a significant progress but the wild-type CRY2-PHR successfully analyzed by this allow us to see the “real” structure without potential artifacts derived from the mutation. The second NSB paper analyzed the wild-type CRY2-PHR but without study of the tetramer structure. Moreover, the present study reveals more photoresponsive structural changes not previously reported, which is arguably the most important structural knowledge for a photoreceptor protein.

[Response] We immensely appreciate the appraisals from the Reviewer and thank her/his for recommending this work to be published without further delay.

Reviewer #3 (Remarks to the Author):

The authors adequately addressed some of the concerns that I raised based on the manuscript originally submitted to Nature Communications. However, I still feel that the conclusions about the CRY photoactivation mechanism, that are drawn based on the presented structure of the light-state Arabidopsis thaliana (At) CRY2 tetramer structure in absence of any functional validation, are rather limited.

This reviewer thinks, that the main value/novelty of the At-CRY2-PHR structure compared to the already published tetrameric bioactive Zea mays CRY1 mutant structure (Shao et al, NSMB 2020, Ref.19) is that it allows a direct comparison of active At-CRY2-PHR with the published (inactive) monomeric At-CRY2-PHR and BIC-inhibited At-CRY2-PHR (Ma et al, NSMB 2020, Ref 37). I.e. potential differences between the activation mechanism of tetrameric Zea mays (Zm) CRY1 and At-CRY2 can be worked out and are now no-longer affecting the comparison between active and inactive CRY2. This is in my opinion not sufficiently clear in the current manuscript. The authors cite the paper reporting the tetrameric Zm-CRY1(W374A) structure for the “proposed photoactivation mechanism in CRYs (intro line 87 to 89)”, but they do not say, what mechanism was proposed for CRY1 and how this differs from the mechanism proposed in this manuscript for At-CRY2. What specifically is new in this manuscript regarding the CRY2 photoactivation mechanism compared to what we already know from the tetrameric Zm-CRY1 structure? The authors partly address this issue structurally in the text and by adding a new table S1, that lists structural differences to the inactive At-CRY2 structures as well as to the active Zm-CRY1 structure, but the functional/mechanistic comparison between At-CRY2 and Zm-CRY1 is still largely missing.

[Response-1] We thank the reviewer for her/his insights. The recent study (Shao et al 2020) reported the first structure of photoactivated mutant CRY1 in maize, the proposed mechanism suggested light-mediated oligomerization that also release the CCT (even though it was not

captured in any structure thus far), and this state allows the downstream interactions. However, without the native structure of CRY2, the previous work was limited and miss an important link to provide compelling evidence for cryptochrome mode of action. Moreover, the mutant CRY1^{W368A} (that may introduce potential unwanted artifacts) is presumably photochemically compromised in photoreduction, yet constitutively active in planta. Nonetheless, the reported oligomeric structure of maize CRY1 (Shao et al 2020) is indeed a major leap in the field, yet our work reveals the structure of the first non-mutated native Arabidopsis CRY2 in a photo-oligomeric state. Moreover, our study not only offers the first comprehensive structural comparison between all plant cryptochromes structures, but also revealed new structural elements and critical residues that are likely to partake in photo-induced oligomerization. We provide the first comparative structural examination of CRYs conformations and recapitulates an intermediate state of electron transport *via* the Trp-triad and offer an important updated model for CRYs photoactivation. We agree with the reviewer that some functional implications of the new structural data reported here await further studies *in vivo*, something that we are very excited to follow up on in the future but are not the in scope of the current study.

[Response-2] We thank the reviewer for pointing out some unclear statements in the text and we have now clarified several points accordingly both in the Introduction, Results, Supplementary information and in the Discussion as can be seen below.

Here are some examples, where the structural comparison needs more work:

We thank the reviewer for the following constructive points, we have now addressed the following point-by-point:

Tetramer Architecture. Lane 140 – 149, 163:

The CRY2-PHR tetramer has the same architecture as the recently published bioactive tetrameric Zea mays CRY1. The 3 Å rmsd deviation that the authors report, appears to be due to the rotation of the C/D molecules compared to A/B (which the authors don't mention, but fig. S3B suggests). Minor differences are observed in the H-T tetramer interface, i.e. a unique 310 helix is not present in Zm-CRY1 and inactive CRY2, instead these have three η -helices. Unfortunately no functional implications of these changes apart from higher flexibility in the CRY2 tetramer are discussed

[Response] We thank the reviewer for bringing this to our attention. The larger deviation observed between the mutant maize CRY1 and Arabidopsis CRY2 reported in our study represents the overall changes in all four chains (A-D) and not only for the rotation of C/D molecules. Moreover, the molecules A/B and molecules C/D represent the active dimer conformation in the tetramer. To better clarify this point we have now included and incorporated another panel into Figure S3b that exemplify the exact deviation between A and B molecules. We have also clarified this point under Results and underlined the structural plasticity of CRY2 compared to the ZmCRY1 structure. We agree with the reviewer that the notion in our text that refers to the unique 310 helix (absent in ZmCRY1 and inactive CRY2 that have three η -helices instead) does not provide further functional reasoning. To address this, we have now clarified this point under Results, and suggested that this flexibility may help to coordinate and fine-tune the dimerization interface during photo-oligomerization process:

This plasticity may result in larger conformational changes during the oligomerization process of CRY2 and can explain the movement of the subdomains that results in fewer interactions within the H-H interface. Also, AtCRY2-PHR_{monomer} and the photo-active mutant ZmCRY1C^{W368A} have three η -helices located in the H-T interface that are completely absent in AtCRY2-PHR_{tetramer}, suggesting more flexibility of the H-T interface and a possible role in fine-tuning the proper orientation of the dimeric interface within the active tetrameric structure (**Figure S3d_(iii)**). This analysis also further corroborates the recent findings that place the H-T interface as the initial interaction surface for inhibition of photo-activation by BIC2, and substantiates this interface as a unique feature of active tetrameric structures^{19,35,36,40}. Further assumptions to address this structural flexibility could be discussed but would be very hypothetical and challenging to test. Yet, we believe that many of the interesting structural elements that we have revealed in this work would be a subject for further biological experiments in future studies.

Interfaces, Lane 149 to 163

The authors compare the CRY2 H-H and CRY2 H-T interfaces of the tetramer with the contacts of monomeric AtCRY2 to its crystallographic symmetry mates. The argumentation and Fig. S3d/e are a bit confusing, but it seems they want to say that the H-T interface in the At-CRY2 tetramer differs more from symmetry mate contacts in the At-CRY2 monomer crystals than the H-H interface. This implies that a similar tetramer was observed in the At-CRY2 monomer crystals by symmetry relationships. The authors should give more background information on the At-CRY2 monomer crystals (space group, symmetry relations etc.), if they want to do this comparison. Additionally, I find a comparison with crystal dimer interfaces (which are likely influenced by crystal contacts) critical for drawing functional conclusions.

[Response] We thank the reviewer important comment and very helpful suggestion. Indeed, our description of the interfaces is more convoluted than it should be. The reviewer's understanding of our comprehensive comparative analysis is correct, and we absolutely agree that this part requires more clarifications. To address this, we have edited, rephrased and clarified this entire part in the results section. We have also changed the color scheme within Figure S3, in particular S3d. We have included an additional panels for figure S3d [(i),(ii) and (iii)] and distinguished between the copies within asymmetrical unit (now in grey color and denoted as copy A' and copy B') of the monomeric structure and the crystallographic two-fold symmetrical mates (now in dark grey and denoted CRY2' symmetry mate). We have included the crystallographic details of the monomer in the Results, in the Figure S3, and provide more details (including the crystal space group) in the figure legend. We are confident that these clarifications and modifications will help the readers to navigate between the different structures. These are some of the main points that are now integrated into the main text and further explain our analysis: *The superposition analysis of AtCRY2-PHR_{tetramer} H-H (monomers A-D and B-C) versus the two copies in asymmetrical unit of AtCRY2-PHR_{monomer} show no major changes in the overall structure (Figure S3d_(i)). However, superposition of AtCRY2-PHR_{tetramer} H-T (monomers A-B) of AtCRY2-PHR_{tetramer} with the two copies of AtCRY2-PHR_{monomer} shows that only monomer A of the tetramer is aligned with the copy A' of the AtCRY2-PHR_{monomer} (Figure S3d_(ii)). Moreover, examination of the crystallographic two-fold symmetry related copies of AtCRY2-PHR_{monomer} shows larger deviation in particularly within the helices that participate in the active dimerization interface (Figure S3d_(iii)). This analysis strongly suggests that the H-H interface region of AtCRY2-PHR_{tetramer} is found to be similar in AtCRY2-PHR_{monomer} and AtCRY2-PHR_{inactive} structures, however H-T interface in AtCRY2-PHR_{tetramer} is distinct during the active oligomeric state. Further comparison of the H-H interface shows fewer hydrogen bonds in AtCRY2-PHR_{tetramer}*

compared to AtCRY2-PHR_{monomer}. This plasticity may result in larger conformational changes during the oligomerization process of CRY2 and can explain the movement of the subdomains that results in fewer interactions within the H-H interface.

Rotamers in Interfaces, Lane 169:

The authors describe 54 rotamer changes between the CRY2-tetramer and monomer (Fig. S3f). Can all of these rotamer changes be seen based on an electron density obtained at 3.25 Å resolution? For some of them (e.g. R439, R208) I doubt it. Please provide electron density figures to support these rotamer changes.

[Response] We thank the reviewer for pointing out this notion. Despite the overall 3.25Å resolution of the structure, many areas within the structure exhibit high ordered and higher local resolution with very well-defined electron density. Nonetheless, we agree that additional electron density will solidify our statement. To address this, we have now added a new panel into Figure S3e (S3e_(ii), also see below) that shows the electron density for representative rotamers residues within the oligomeric interface.

Connector Loop, Lane 180 onwards:

The authors describe conformational changes in the connector loop between the active tetrameric AtCRY2 and ZmCRY1 structures compared to the inactive monomeric CRY2 and BIC inhibited CRY2 structures and report enhanced flexibility in the active structures. They suggest movement of the N- and C-terminal CRY domain (lane 191), which is not obvious based on the presented structural comparisons. Please clarify. moreover, conformational heterogeneity in the connector loop is frequently observed between different CRYs, as it is a flexible and dynamic region, and the significance of these changes for photoactivation is not clear based on the current manuscript. Please clarify.

[Response] We agree with the reviewer that the connector loop is a flexible and dynamic region observed in all the CRY structures determined so far. Nonetheless, the connecting loop has been highly conserved among all CRYs and has been implied to have a biological function (see below). Our crystal structure together with previously determined models provide a unique opportunity to compare and analyze the conformational heterogeneity between the active and inactive cryptochromes. To our knowledge this study is the first to project the importance of the connector loop and highlight the differences in these regions between the inactive CRY2 and active CRY2. While the biological significance of this loop remains to be fully elucidated *in vivo*, our analysis suggests that $\alpha 6$ dynamics may play role in the oligomerization process. This assumption is now further discussed under Discussion and underlined the recent *in vivo* studies of photoexcited Arabidopsis CRY2 that identified four key residues (T149, S162, T178, and

S188) that undergo phosphorylation upon photoexcitation (Liu, 2017, Nature Comms). All of these residues are positioned within the connecting loop and the short helices of CRYs (T178, S188). Thus, it is possible that other modifications within this connecting loop may affect its overall dynamics and photo-oligomerization process.

To better illustrate the movement of the connecting loop between active and non-active state, we have included in the submission two short videos that highlight the structural conformations of the connecting loop of tetrameric CRY2 in comparison with ZmCRY1 and monomeric CRY2.

Inhibition of CRYs by BIC (from lane 207)

The authors compare the H-T dimerization interface in the At-CRY2 tetramer with competing CRY2-BIC interactions. Was the competition of BIC with the H-T tetramer interface not evident based on the Zm-CRY1 tetramer structure? What is new based on the At-CRY2 tetramer structure?

Lane 222: The At-CRY2 structure “reveals key residues E462, T465, N466.....” The authors should say if these residues conserved in Zm-CRY1 tetramer or different ?

[Response] In this work we have thoroughly examined and compared all the structural elements that were reported here and previous works and may involve in BIC-CRY interfaces in our CRY2 structure (shown Figure 2, 4 and Table S1). The tetrameric structure of CRY2 reported here is adding yet more layers of understanding of how native cryptochrome-2 oligomerization can be affected by BIC interaction.

Specifically, we found that the interaction between the monomers (A-B, H-T interface) in AtCRY2_{tetramer} and ZmCRY1C (monomers A-D, H-T interface) is slightly different, hence the structure of AtCRY2_{tetramer} is crucial for underlining all residues that involved in the disruption of dimerization interface upon BIC binding.

Furthermore, our structural analysis not only corroborates certain residues involved in the CRY2-BIC interface as suggested previously, but also reveals new key conserved residues such as E462, T465 and N466 that are likely involved in oligomerization and the BIC2 mechanism of inhibition (Figure 4b-c).

We agree that the conservation of these residues was not clear in the main text and may lead the readers to scroll to the alignment map that provided in Figure S1. Therefore, to clarified this notion we edit the text under Results and included an additional sentence under Discussions:

“Detailed structure-based analyses using the structure of AtCRY2-PHR_{tetramer} allowed us to also corroborate the recently reported CRY2 and CRY1 structures (Ma et al., 2020; Shao et al., 2020), and further to reveal novel residues that are involved in CRYs oligomerization as well. The newly identified residues E462 and N466 are highly conserved among all CRYs, and T465 in CRY2_{tetramer} is substituted to serine residue in all CRY1s (Figure S1).”

FAD binding pocket:

Lane 252: “D359 (ZmCRY1) vs N345 (At-CRY2) explain reported differences in light sensing mechanism between CRY1 and CRY2.” The authors should explain which “differences in light sensing mechanism between CRY1 and CRY2” exactly they are addressing and provide a mechanistic rational why they think that D359 vs. N345 affects exactly these light sensing differences.

[Response] We agree with the reviewer that this statement is ambiguous and not provide sufficient information regarding the distinct mechanistic rational. To address this, we have rephrased and clarified this paragraph under Results as follows:

“D359 may play an additional role in donating a proton to the flavin moiety as was reported previously (Kottke et al., 2006), and both D359 (CRY1s) and N356 (CRY2s) are positioned within the ATP binding site of CRYs (Eckel et al., 2018; Engelhard et al., 2014; Ma et al., 2020). The importance of this substitution and its potential involvement in the distinct photosensitivity that was reported between CRY1 and CRY2 (Lin et al., 1995a; Lin et al., 1995b), await further research.”

Lane 256/257: 180° rotation in FAD O3 and O4. I doubt that this can be derived from a 3.25 Å resolution map. The authors should show this region with electron density, to show how the electron density defines this 180° rotation.

[Response] We thank the reviewer for pointing out the notion regarding the FAD density. We would like to emphasize that the 3.2Å resolution reflects the overall resolution of the entire tetrameric structure. The residues and FAD-bound in the core region of CRY2 have tightly packed and highly ordered atoms arrangements, this result in a very well-defined electron density, despite the overall resolution. There are numerous crystal structures of macromolecules that have much higher overall resolution but contain less ordered portions that result in poorer local resolution. This is not the case for the CRY2 structure reported here, in particular with FAD. To clarify this notion, we have now replaced Figure S4a-b with a better color scheme of the electron density of FAD (now in S4b) and surrounding residues (now in S4a) so the electron densities will be sharper and visible. Nonetheless, we have also provided in this letter for the Reviewer, all the electron density maps of FAD for each chain (2foc, sigma=1, see below).

Importantly, after inspection of the electron density of CRY2-BIC2 previously reported structure in NSMB (Ma et al, 2020), we found that atoms O3 and O4 of FAD has been mistakenly placed by the authors (Ma et al, 2020) in a wrong *cis* configuration and hence the difference observed between CRY2_{tetramer} and CRY2_{inactive}. After identifying and verifying this error, we have removed all indications that relate to the 180-degree flip that can be misleading and incorrect (this was also corrected in Figure S4). Nevertheless, all our analysis of the FAD pocket and all related conclusions that stem from our crystal structure remain the same.

Trp triad:

Lane 275: “atomic resolution” is a bit bold for only 3.25 Å resolution.

The authors find that the distance between the middle W374 and the distal W321 is larger (6.5 Å) in the active *At*-CRY2 than in inactive *At*CRY2 monomer (4.8/4.9 Å), and also somewhat larger than in the also tetrameric active *Zm*-CRY1 (5.9 Å).

The distances W374-W397 and W397-D393 are shorter in the active At-CRY2 tetramer compared to inactive At-CRY2 and also compared to active Zm-CRY1. This is confusing, as here the active Zm-CRY1 resembles inactive At-CRY2 and is different from active At-CRY2. The authors should explain this. D393 – N5 FAD: here active Zm-CRY1 and At-CRY2 both have shorter distances than inactive At-CRY2. Here tetrameric active Zm-CRY1 seems more similar to tetrameric active At-CRY2.

Is the Trp triad and D393 arrangement of tetrameric Zm-CRY1 more similar to tetrameric active At-CRY2 or to inactive At-CRY2? Please clarify this and potentially discuss mechanistic differences for the active CRY1 and active CRY2 state.

[Response] We thank the reviewer for pointing this out. We have now removed 'atomic resolution' from this sentence.

In regard to the distances of Trp-triad - in all the crystal structure of CRYs such as AtCRY1, AtCRY2_{monomer} and ZmCRY1C^{W368A}, the distances between tryptophan residues gradually increase as it moves farther away from the FAD moiety. We believe that the source of the confusion is related to the fact that ZmCRY1C^{W368A} is mutated in middle Trp triad to Ala. Thus, the variation in the distances between AtCRY2_{tetramer} (W374-W397) reported here and ZmCRY1C^{W368A} (A368-W391, Shao et al 2020) is mainly because the later structure is mutated and exhibits larger distance between the middle and proximal Trp residues.

In the cryptochrome field there is a large debate regarding the significance of the Trp-triad and their role in photoreduction *in vivo*. The mutation in the middle Trp (W368A in CRY1, W374A in CRY2) shows distinct photoactivities between *in vitro* and *in planta* studies. *In vitro*, CRY2 with W374A was reported to have only residual photoreduction activity, yet *in planta* the mutant exhibits a constitutively active phenotype. While our analysis cannot fully settle this debate, we believe it is important to provide a thorough structural observation and analysis that compares between the different photoactivated states and underlines the photoactivated CRY2 determined in this work.

To better clarify this point, we have now added the following paragraph under Discussion:

*Although both photoactive CRY structures share a similar N5(FAD)-D393 distance and relatively shorter D393-W397 distance, the precise photoreduction mechanism of the W368A mutant that was used in our comparative analysis, remains to be fully elucidated. Previous studies *in vitro* have shown that W374A mutation in CRY2 results in only residual photoreduction activity which may explain the larger Trp-triad distances found in our analysis (Li et al, 2015, Gao et al 2015). Yet, *in planta* W374A mutant results in a constitutively active phenotype, thus it is also possible that alternative photoreduction pathway(s) can contribute to the photoactivated states as suggested previously.*

Overall the presented structural differences between active/inactive CRY2 and active CRY1 do not convincingly show, which features are characteristic of photoactivated At-CRY2 and why. The discussion unfortunately only recapitulates the observed structural changes, but also does not discuss clear mechanistic implications for light-activation derived from these observation.

I therefore think that this manuscript is more suited for a journal that is specialized on structural biology.

[Response] All suggestions and the requested additional data are now well implemented into the text and we thank the review for her/his help to improve the manuscript and its clarity. Nature Communications and the emerging Communications Biology Journals are home to numerous exciting and groundbreaking structural biology studies. Here, we determine the

crystal structure of active plant blue-light photoreceptor CRY2 and provides for the first time detailed structural insights into the photoactivation of plant cryptochromes. We are confident that this work will have impact not only in structural biology of light sensors but also in plant biology and photobiology fields.

REVIEWERS' COMMENTS:

Reviewer #3:

Remarks to the Author:

The authors have now addressed all my concerns.

I recommend publication as is.